# Indications of food insecurity in the content of telephone calls to a community referral system

**Nasser Sharareh**[1]*, **Ching-Yu Wang**[2], **Andrea S. Wallace**[2], **Jorie Butler**[3,4]

**1** Department of Population Health Sciences, University of Utah Spencer Fox Eccles School of Medicine, Salt Lake City, Utah, **2** College of Nursing, Salt Lake City, Utah, **3** Departments of Biomedical Informatics, University of Utah Spencer Fox Eccles School of Medicine, Salt Lake City, Utah, **4** Department of Internal Medicine, Division of Geriatrics, University of Utah Spencer Fox Eccles School of Medicine, Salt Lake City, Utah

☯ These authors contributed equally to this work.

* Nasser.sharareh@hsc.utah.edu

**Data Availability Statement:** The data relevant to this study are owned by a third-party organization, United Way of Salt Lake and cannot be shared unless the requester signs a data-sharing agreement with this organization. Below, please

## Abstract

### Background

Food insecurity is a social determinant of health that impacts more than 10% of U.S. households every year. Many unexpected events make food-insecure people and those with unmet food needs seek information and help from both formal (e.g., community organizations) and informal (e.g., family/friends) resources. Food-related information seeking through telephone calls to a community referral system—211 network—has been used as a proxy for food insecurity but the context of these calls has not been characterized and the validity of this proxy measure is unknown.

### Objective

To investigate the content of food-related telephone calls to 211 and explore the indications of food insecurity during these calls.

### Methods

We conducted a secondary qualitative analysis on the transcripts of food-related calls to Utah's 211. From February to March 2022, 25 calls were sampled based on the location of callers to ensure the representation of rural residents. 13 calls from metropolitan and 12 calls from nonmetropolitan ZIP Codes were included. Using a purposive sampling approach, we also made sure that the sample varied with regard to race and ethnicity. Calls were transcribed and de-identified by our community partner—Utah's 211 and were analyzed using a thematic analysis approach by our research team.

### Results

Three themes emerged from the qualitative analysis including referral to 211, reasons for food-related calls, and reasons for unmet food needs. Results highlight the complex social

find their contact information: Sandra Carpio, 211 Managing Director sandra.carpio@uw.org 257 East 200 South, Suite 300 | Salt Lake City, UT 84111-2078 | tel 801.259.3139 | fax 801.736.7800 | www.uw.org.

**Funding:** NS Health System Innovation and Research (HSIR) 2022 University of Utah Pilot Project Award https://medicine.utah.edu/population-health-sciences/divisions/hsir The funders had no role in study design, data collection and analysis, decision to publish, or preparation of the manuscript.

**Competing interests:** The authors have declared that no competing interests exist.

environment around 211 food-related callers, lack of knowledge about available food resources, and indications of food insecurity in calls.

## Conclusion

Information seeking for food-related resources through 211 is a problem-solving source for people living in a complex social environment. Indications of food insecurity through these calls validate the use of these calls as a proxy measure for food insecurity. Interventions should be designed to increase awareness about the available resources and address the co-existing social needs with food insecurity.

## Introduction

Food insecurity has impacted more than 10% of U.S. households every year over the past two decades [1]. People can experience food insecurity regularly or occasionally throughout a year [1]. While lack of income and financial resources are considered the main risk factors for food insecurity, there are many other factors and situations placing people at risk of food insecurity. For example, natural disasters can disrupt the food supply chain and create local surges of food insecurity [2, 3], among which, hurricanes, wildfires, and earthquakes are common in the U.S. COVID-19 caused business closures and food shortages that led to intermittent and local surges in food insecurity [4–6], hence, future pandemics could generate the same problems. Lack of donations to food pantries [7] and the increasing rate of inflation [8] make it harder for food pantries to meet the demand and drive them to closure, which will put people who rely on emergency food resources at higher risk of food insecurity. This is an important consideration as not every food pantry user is food insecure [9] and limiting their access to food might drive them to food insecurity. These varied, unexpected circumstances make people with unmet food needs and at risk of food insecurity reach for help from other resources. Information seeking for food-related resources is a complex phenomenon that has not been studied in depth. Some people with prior experience seek help from formal food resources (e.g., food pantries, Supplemental Nutrition Assistance Program (SNAP) benefits), and some seek help from informal resources such as family and/or friends. However, many others have limited knowledge about how to meet their unmet food needs and seek assistance from community organizations.

The 211 network has the most comprehensive source of information about local community resources [10], helping people across the U.S. and Canada by providing information and referrals to social assistance programs. There are more than 200 211 agencies across the U.S. (available to more than 300 million people), each with a team of highly trained community resource specialists who connect callers to free and low-cost community resources for needs like food, housing, and healthcare [10]. 211 services are funded by the states and donations and are operated by different organizations such as United Way—a privately funded charity, actively working in 95% of U.S. communities to improve education, economy, and health—[11] or local crisis centers, all providing services in 180 languages using non-English operators and access to a translation line. By calling 211, callers will be connected to community resources to address their unmet needs for housing, food, and many other social needs. At the time of the call, a 211 resource specialist documents callers' ZIP Code, gender, primary language, and reason for the call. Race/ethnicity, however, is not uniformly collected. The reason for the call is identified using a taxonomy. For instance, a call would be flagged as a food-

related call if there was any indication of food needs during the call such as requesting information on how to access food pantries or soup kitchens or help with enrolling in government nutritional assistance programs. Callers may also share their contact information, so that community organizations can reach out to them

211 food-related calls have been used to inform the allocation of resources to areas with unmet food needs [12–15]. Researchers have also used the real-time, ZIP-Code level food-related calls to 211 as a proxy for food insecurity [12–15] as food insecurity estimates are only available at the national and state level and are reported annually, while 211 can report ZIP Code-level data daily. However, the validity of this proxy measure has not been evaluated and this limitation has been documented in the literature [13, 16]. Moreover, the information seeking behavior and the content of these calls are largely unknown. Transcripts of food-relate calls to 211 could provide in-depth information about the reasons for calls and can be assessed against common food insecurity measures. As such, our goal of this study was to qualitatively characterize the nature of 211 calls for food resources and assess whether and how these calls indicate food insecurity.

## Material and methods

This paper stemmed from a unique community partnership that the lead author initiated with the United Way of Salt Lake, which operates the 211 services in Utah. The Institutional Review Board at the University of Utah has approved this project and waived the need for consent. Methods for data collection, analysis, and reporting were guided by the Consolidated Criteria for Reporting Qualitative Research (COREQ) guidelines [17].

We conducted a secondary qualitative analysis of existing administrative data. The existing data were audio recordings of food-related calls to Utah's 211. The United Way of Salt Lake keeps audio recordings for a month from the time of the call. While we did not have access to the audio recordings, we provided the United Way of Salt Lake with a sampling algorithm, with one main criteria, to identify 25 food-related calls and transcribe them for our research team. We initially requested 25 transcripts as it is a reasonable sample size for qualitative analysis [18, 19]. Our main criterion was to receive transcripts of 13 calls from metropolitan and 12 calls from nonmetropolitan ZIP Codes to ensure representation of rural residents as they might have different food insecurity experiences. In Utah, 47% of ZIP Codes are nonmetropolitan areas according to the rural-urban commuting area codes [20]. Moreover, using a purposive sampling approach [21], we requested a sample that varied in regard to race and ethnicity similar to the Utah population. 77% of Utah's population are non-Hispanic white and 14.4% of the population are Hispanic or Latino [22]. Considering that Utah 211 collects race/ethnicity in the same field, non-Hispanic could be white or any other race and non-white may be Hispanic. We also did not restrict our sample to English-speaking callers, but we do not know whether those callers used a non-English operator or a translation line as we were only provided the transcript of calls.

From February to March 2022, the 211 senior community resource specialist identified the food-related calls that matched our criteria upon receiving them, transcribed the calls within the next few days, and provided us with 25 de-identified transcripts. While we did not select those calls, we designed the sampling criteria and made sure that our final sample was aligned with those criteria. In addition, while each call can be classified into more than one category based on the discussion during the call (e.g., food, housing, healthcare), our focus in this paper was on calls that had indications of food needs. Meaning that other needs, besides food-related needs, could have been discussed during these selected calls.

## Qualitative analysis approach

We used thematic analysis [23, 24] to analyze our unconventional but novel qualitative data—transcripts of food-related calls. Thematic analysis is a powerful approach to determining a set of experiences, thoughts, or behaviors across qualitative data [25] and provides a well-structured approach to assessing the perspectives of research participants and highlighting their similarities and differences [26]. This approach organizes qualitative data into codes and themes, similar to other qualitative methodologies such as grounded theory and ethnography [25]. Thematic analysis is also widely used in qualitative analysis and in health-related fields [27] and works well at bringing together multidisciplinary teams [24].

We conducted the thematic analysis using ATLAS.ti software (version 22 Windows [28]). Our qualitative analysis team comprises one food insecurity/health services research expert (NS, Ph.D.), one behavioral scientist with extensive experience in qualitative analysis (JB, Ph. D.), and one Ph.D. student focused on qualitative analysis (CW, MSN, RN). None of the callers were involved in our analysis or in the interpretation of results. We followed the 6 phases of thematic analysis approach (Table 1). In phase 1, we reviewed the 25 transcripts to become familiar with the data and to note potential codes and themes. During this phase, we spent time as a group discussing the strategy of data management and analysis to make sure each phase will execute effectively. In phase 2, we gathered as a group to code parts of the transcripts and created an initial code book. After the group coding sessions, CW and NS coded the remaining transcripts according to the agreement in the group coding session. JB reviewed all the codes. We resolved disagreements in code names or coded quotes through discussion and coder consensus. Our initial codebook included code names, definitions, and sample quotes. During phase 3, similar codes are grouped together to represent preliminary themes. We used ATLAS.ti to create a diagram to illustrate and organize codes into themes that richly characterize our qualitative data through a series of group discussions. In phase 4, each identified theme was evaluated by our research team. We modified, merged, and deleted themes, so that the final themes were accurate and meaningful and did not overlap with each other. In this phase, we also concluded that thematic saturation—the point at which transcripts do not reveal new information [29]—was reached using the sample and, thus, no additional transcripts were sought for analysis. In phase 5, we identified 3 final themes with a description of code groupings, sample quotes, and theme description. Finally, in Phase 6, we developed this manuscript and revised the content based on the reviewers' feedback. Note that thematic analysis is an iterative process in which each phase could inform us to revise the prior phases [26].

**Table 1. Steps of our qualitative data analysis.**

| Phase | Actions |
| --- | --- |
| **Phase 1: Familiarizing the transcripts** | Read the transcripts<br>Took notes for codes and themes<br>Established a strategy for data analysis and data management |
| **Phase 2: Creating an initial code book** | Discussed the codes<br>Coded the transcripts |
| **Phase 3: Discovering themes** | Created a diagram to illustrate codes<br>Discussed the potential themes list |
| **Phase 4: Reviewing the themes** | Reviewed the themes<br>Evaluated each theme<br>Assessed thematic saturation |
| **Phase 5: Finalizing the themes** | Gave each theme a name and definition |
| **Phase 6: Preparing reports/manuscripts** | Developed/revised this manuscript |

## Results

A total of 25 transcripts of food-related telephone calls, 13 from metropolitan and 12 from nonmetropolitan ZIP Codes, were analyzed. Transcripts had an average of 961 words with a range of 173–4,081. Table 2 shows the descriptive statistics of our sample. The reason for missing data for some variables is that this information is not uniformly gathered by the 211 specialist at the time of a call, and even sometimes, the information was inferred from the contents of the calls.

Through our qualitative analysis, three themes merged from the transcripts including 1) referral to 211; 2) reasons for food-related calls; 3) reasons for unmet food needs. Each theme is reported below along with representative quotes. The theme with the highest number of coded quotes was reasons for unmet food needs followed by referral to 211. The full list of quotes is available by request from the authors only if the requester signs a Data Sharing Agreement with the United Way of Salt Lake.

### Theme 1: Referral to 211

Participants sought out food resources by contacting the community helpline 211. Throughout the transcripts, there were many indications of how and where callers heard about 211 either through social media, family and friends, or other community organizations.

*Watching YouTube and because of all of the things with COVID, there's a guy that I watched that's a financial advisor and he tells everybody, you know he was like "there are still programs out there If you haven't gotten this. If you need help with rent, you go to 2-1-1."*

*Agent: how did you hear about us?*

*Caller: Facebook*

*Agent: OK, so how did you hear about us?*

*Caller: Through the radio.*

*I was recommended your number by an in-law.*

*Agent: It is, OK? So how did you hear about 211?*

*Caller: Uh, my girlfriend's on it.*

*They're closing the food bank and they just referred me.*

*They told me about 211 through. . . When I called the Food bank warehouse.*

**Table 2. Descriptive statistics of 25 selected food-related telephone calls to Utah's 211.**

| Variable (number of people reported) | Description |
|---|---|
| **Age (8)** | Mean: 35.5 –Median: 32.5 –Range: 19–58 |
| **Household size (13)** | Mean: 3.15 –Median: 3 –Range: 1–7 |
| **Education level (13)** | 4 Some college, 7 high school graduate, 1 GED, 1 middle school |
| **Race/ethnicity (25)** | 5 Hispanic; 8 non-Hispanic; 7 white; 5 non-white |
| **SNAP utilization (8)** | Out of 8 callers who responded to this question, 6 were not utilizing SNAP. |

A purposive sampling procedure was used to select those 25 food-related calls.

SNAP: Supplemental Nutrition Assistance Program

*They're closing the two CAP food banks and they just referred me.*

*I heard about you through the SNAP benefit program.*

## Theme 2: Reasons for food-related calls

As expected, many callers stated that their reason for calling was lack of food and limited knowledge about how to meet their unmet food needs (i.e., informational barrier). While some knew about food pantries and the SNAP program, they had limited knowledge about the location of nearby food resources and their operating hours or did not know how to enroll in SNAP.

*I just heard about this yesterday. I didn't even know this was a thing but, I uhm, I am in desperate need of some food for my family.*

*I have no food.*

*I was told that I can call this number to find help with food.*

*If I could just get some food, I'd be OK.*

*Agent: let me ask you this before I do put you on hold, do you currently have food in the house?*

*Caller: No, uhm, not much.*

*I was wanting to ask where is a food bank near me.*

*We've gone door to door. Nobody knows where there's a food bank in Pleasant Grove or American fork. And the closest one I found was in Riverton and my, anyway is there any in Pleasant Grove or in American work that. . .*

*Do you know any food pantries that are open on the weekends in Grantsville? Tooele or Grantsville area?*

*Wondering if I can get any information on food pantries nearby.*

*I'm trying to see if there are any food banks that are open in my area today or not, maybe not even in my area, some that are open just period.*

*Can you tell me what food banks are open near me?*

*I was just trying to locate any nearby food banks or food distribution and the closest one to me so I can go to them.*

*I was told to call and ask about the food assistance and, what places and times would I be able to do that?*

*I don't know where to start.*

*How do I fill out for money for food stamps?*

## Theme 3: Reasons of unmet food needs

Many risk factors cause unmet food needs including limited money to buy food, ineligibility to enroll in SNAP, and other co-existing social needs.

Lack of money to buy food, the main risk factor of food insecurity and the main criteria of USDA's food insecurity measurement, was indicated by some callers.

*I only receive $403.00 a month.*

*We've gone through every cent of savings.*

*I lost all my money this month.*

*We didn't even make half of what we made last year.*

In our sample, some experienced difficulties accessing governmental resources because their income was slightly higher than the requirements.

*I was trying to get food assistance from SNAP, and they told me I didn't qualify by $40*

*I called SNAP and they said that I made a few dollars over the amount*

*Agent*: *Are you receiving any assistance from the government like Medicaid or Medicare*?

*Caller*: *No, we make $3 too much.*

Some of the callers indicated difficulties to pay for food because they were dealing with other co-existing social needs (e.g., healthcare, rising costs, unemployment, family caregiving, lack of transportation) that put them in a closed loop of problems hard to address. These competing financial demands limit the allocation of funds to address unmet food needs.

*I have all of our medical covered, but we're having a struggle with food*

*I've been in and out of the hospital since before Thanksgiving. They're sending me paperwork so that I can get a discount to see if they can write some of the things off.*

*I have so many medical bills because I got really sick, and I've been sick since before Thanksgiving and have been in and out of the emergency room and have had home health care coming and stuff like that so.*

*COVID is killing us. We have lost six people so we're not doing very well mentally.*

*All the gas prices and everything going up and uhm—Because groceries are killing us, like we're picking and choosing between medicine and food.*

*Like 'cause we've been OK it's just the food and the thing that has gone up in our life now has been the rent.*

*Is there anything that can help me with my school or help me with paying bills or any school loans or paying bills.*

*I was paying $10 for a prescription, and now it's $80.00. I had another medicine that's for migraine and chronic pain and it was $2.41. And now it's $42. So, I'm like, that's why I'm saying, like medicine costs have gone up.*

*We're all roofers and we haven't been working this winter cause of the snow. So it's been really hard.*

In addition, taking care of their family was one of the main reasons that caused unmet food needs.

*We take care of my elderly mother who's fallen and then in and out of the hospital it's been. It's been rough and like I said I lost both my uncles. And then I just lost my aunt the day before my birthday.*

*I got desperate when my daughter-in-law called me, and she was like I have $8.50, and I haven't bought groceries and I'm like, I don't have any money to give you*

*I need a lot of help. So, I'm calling for two families. Myself and my daughter in law. And so, uh, my husband, our (household) We didn't even make half of what we made last year and then my son tried to commit suicide in November, and he hasn't been working and he's autistic. He is on the spectrum and stuff.*

*I would like to find something to help me assist my wife and I, she had a brain aneurysm, and I've been taking care of her, and I took an early retirement to do that*

Some barriers were physical such as challenges with transportation both in urban and rural areas.

*Caller*: *I don't drive.*

*Agent*: *So, you don't have a car?*

*Caller*: *No.*

*I need to find out if any of them deliver cause I don't have a vehicle*

## Discussion

In this study, we conducted a secondary qualitative analysis of transcripts of 325 food-related telephone calls to a community referral system—211 network—to investigate the content of those calls and explore the indications of food insecurity in them. Through our analysis three main themes were emerged: referral to 211, reasons for food-related calls, and reasons for unmet food needs.<

Information seeking for food-related resources was initiated due to the lack of food and information on how to address unmet food needs. There is also evidence that people have limited information about available resources, their locations, and operating hours. People have unmet food needs and are either food insecure or at risk of food insecurity and do not know where and how they can address those needs. While increasing the availability of emergency food resources such as food pantries could increase access to food [30], as long as people do not hear about them, these resources cannot efficiently reduce unmet food needs. This finding is consistent with the literature, documenting that 47% of low-income people do not use food pantries due to informational barriers (i.e., lack of information) [31]. Analyzing the transcripts also highlighted the fact that some people are not using government assistance programs like SNAP (see Table 2) but they still have unmet food needs. Our findings are also supported by the 2020 national estimates as only 36.5% of food-insecure households used food pantries [9] and among eligible households that did not enroll in SNAP, 24.8% were food-insecure [1]. Therefore, there is a critical need for educational campaigns to increase awareness about available food resources.

There was evidence of food insecurity during calls. A few callers directly referenced a lack of money to buy food. However, our analysis revealed additional evidence of a lack of money due to other co-existing social needs. Many callers mentioned that they have paid up their rent, medication, and other healthcare costs and are left with no money and that is why they reached out to 211 to find a way to meet their unmet food needs. This finding emphasizes the usefulness of information seeking for food-related resources through 211 in capturing food insecurity in real time. Food insecurity is currently measured by the U.S. Department of Agriculture (USDA) [1]. However, these estimates are released annually and cannot guide the

timely allocation of resources to communities in need [16]. Considering the indications of food insecurity in 211 food-related calls, the 211 system could be used as a surveillance system to monitor food insecurity spikes at the local level in real time. A real-time surveillance system could also increase the efficiency of our community-level interventions to address food insecurity in a timely manner.

Through our analyses, we also shed light on the complexities of the social environment impacting those calling 211 for food-related needs. People called 211 to meet their food needs but during a call, many other problems were discussed. For instance, the burden of family caregiving was indicated as one of the main problems in a quest for food. Callers lost their job or retired earlier to take care of their beloved ones, which made paying for food harder. It has been established that caregivers are at higher risk of food insecurity as well [32]. Our analysis also highlighted how a multifaceted and complex problem food insecurity is and how it is interconnected with many other co-existing social needs. Callers mentioned that paying for medical bills, unemployment, disability, increasing costs of gas, grocery, and rent, and the impact of COVID-19 on their mental health have created their unmet food needs a complex problem to address. This finding indicates the importance of addressing these overlapping problems together and tailoring our interventions based on people's needs and not just by independently addressing one social need at a time.

The 211 food-related callers also face many barriers in accessing food-related resources. Transportation issues have been mentioned by both rural and urban residents. Ineligibility for SNAP just because of a slightly higher income was as well the reason for some calls and also the reason why people were searching for other resources to address their unmet food needs. One caller also did not know how to apply for SNAP, which again indicates the informational barriers around food resources.

## Limitations and future work

Not many people directly referenced lack of money maybe because the 211 agent did not ask related questions that could lead to income-related answers during the calls. However, there is plenty of evidence of a lack of money due to other social needs such as medications, rent, and unemployment. These indications point to the fact that 211 food-related calls could be used as a proxy for food insecurity; however, future work could consider conducting interviews with food-related 211 callers to measure their food insecurity using established methods such as USDA's food security module or the 2-item Hunger Vital Sign. We also acknowledge that our themes are interrelated but each theme points to distinct facets of food insecurity represented within the call transcripts. In addition, although the transferability of the project findings to other settings is difficult to state, our sample is representative of the Utah's metropolitan/non-metropolitan and racial/ethnic classification and is generalizable as we reached a saturation point of important themes and quotes; however, this project could be replicated in states with more rural population, higher or lower food insecurity rates, and different race and ethnicity diversity.

## Conclusion

Our secondary qualitative analyses of the transcripts of food-related information seeking calls revealed information that could be critical to policymaking and the allocation of food resources. Specifically, we highlighted the relevance of 211 food-related calls to food insecurity and identified the complex social environment and multifaceted barriers that callers are dealing with. Indications of food insecurity through these calls validate the use of these calls as a proxy measure for food insecurity, however, further analysis is needed to measure FI among

these callers using established methods. We also highlighted the need for educational campaigns to increase awareness about available food resources.

## Author Contributions

**Conceptualization:** Nasser Sharareh, Andrea S. Wallace, Jorie Butler.

**Data curation:** Nasser Sharareh, Ching-Yu Wang.

**Formal analysis:** Nasser Sharareh, Ching-Yu Wang, Jorie Butler.

**Funding acquisition:** Nasser Sharareh.

**Investigation:** Nasser Sharareh, Ching-Yu Wang, Jorie Butler.

**Methodology:** Nasser Sharareh, Ching-Yu Wang, Jorie Butler.

**Software:** Nasser Sharareh, Ching-Yu Wang.

**Supervision:** Nasser Sharareh, Andrea S. Wallace, Jorie Butler.

**Validation:** Nasser Sharareh.

**Writing – original draft:** Nasser Sharareh, Ching-Yu Wang, Andrea S. Wallace, Jorie Butler.

**Writing – review & editing:** Nasser Sharareh, Ching-Yu Wang, Andrea S. Wallace, Jorie Butler.

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
