## [Decision Letter · Decision Letter 0]

20 Dec 2022

PONE-D-22-22201Real-time indications of food insecurity in telephone calls to a community referral systemPLOS ONE

Dear Dr. Sharareh,

Thank you for submitting your manuscript to PLOS ONE. After careful consideration, we feel that it has merit but does not fully meet PLOS ONE’s publication criteria as it currently stands. Therefore, we invite you to submit a revised version of the manuscript that addresses the points raised during the review process.

 Please submit your revised manuscript by Feb 03 2023 11:59PM. If you will need more time than this to complete your revisions, please reply to this message or contact the journal office at plosone@plos.org. Please include the following items when submitting your revised manuscript:A rebuttal letter that responds to each point raised by the academic editor and reviewer(s). You should upload this letter as a separate file labeled 'Response to Reviewers'.A marked-up copy of your manuscript that highlights changes made to the original version. You should upload this as a separate file labeled 'Revised Manuscript with Track Changes'.An unmarked version of your revised paper without tracked changes. You should upload this as a separate file labeled 'Manuscript'.

We look forward to receiving your revised manuscript.

Kind regards,

Bettye A. Apenteng

Academic Editor

PLOS ONE

Journal Requirements:

2. Please ensure that you have specified (1) whether consent was informed and (2) what type you obtained (for instance, written or verbal, and if verbal, how it was documented and witnessed). If your study included minors, state whether you obtained consent from parents or guardians. If the need for consent was waived by the ethics committee, please include this information.

Reviewers' comments:

Reviewer's Responses to Questions

**Comments to the Author**

1. Is the manuscript technically sound, and do the data support the conclusions?

Reviewer #1: Partly

Reviewer #2: Yes

2. Has the statistical analysis been performed appropriately and rigorously? 

Reviewer #1: N/A

Reviewer #2: N/A

3. Have the authors made all data underlying the findings in their manuscript fully available?

Reviewer #1: Yes

Reviewer #2: Yes

4. Is the manuscript presented in an intelligible fashion and written in standard English?

Reviewer #1: Yes

Reviewer #2: Yes

5. Review Comments to the Author

Reviewer #1: Manuscript PONE-D-22-22201: Real-time indications of food insecurity in telephone calls to a community referral system

While interesting enough, I am not recommending for this paper to be accepted for publication in PLOS One, for two key reasons. First, by reporting on a small study with a specific geographical context, the scope of its contribution is small, and insufficient to merit inclusion in an international journal such as PLOS One. When reworking this paper for submission elsewhere I would encourage the authors to think carefully about how their findings might have relevance beyond the immediate research context and bring this out more clearly in the revised work. Second, I have reservations about the paper’s scholarly rigour, especially the analyses, which identified overlapping analytical themes and thereby felt weak and lacking robustness.

Below I have listed more specific comments on the paper and suggestions for how it could be strengthened and I wish the authors all the best in doing so.

Title

The title ‘Real-time indications of food insecurity in telephone calls to a community referral system’ is not an accurate representation of the focus of this paper, as the paper contains no temporal element (such as how calls to the community referral system have changed over a given time period, or fluctuations over the course of a month). I would like to see the title provide a closer reflection of the paper’s focus and scope.

Introduction

Linking natural disasters to food security (p2) is an important consideration in the Global South but its relevance to the Global North (where this study is situated) is not clear.

The sentence ‘Lack of donations to food pantries [7] and the increasing rate of inflation [8] make it harder for food pantries to meet the demand and drive them to closure, which will put people who rely on emergency food resources at risk of food insecurity’ (p2) misunderstands the nature of food insecurity and the role of foodbanks in assisting those experiencing food insecurity. People who rely on emergency food are almost by definition experiencing food insecurity, and the inadequacy of food banks in alleviating food insecurity is well documented. It is therefore inaccurate to talk of foodbanks as protecting people from food insecurity.

More contextual information is needed about the 211 network (p3). In particular, what is its purpose? Who is it funded by? Does it offer solely information or is it a means of callers being referred to relevant services? And who are United Way? Your international readers may not have heard of this organisation or be familiar with its goals and purpose of operation.

The authors state that ‘Researchers have also used food-related calls to 211 as a proxy for food insecurity’. What is the value for doing so? What can it add to national prevalence estimates, for example? Unpacking this point would help the authors demonstrate the value of this study more convincingly.

Material and Methods

The justification for locating this study in Utah need justifying. Presumably it reflects the researchers’ location but a more convincing reason is needed, alongside reflection on the potential transferability of the project findings to other settings.

Why did the researchers sample 25 calls? What proportion of monthly calls to the 211 helpline is this?

Dates of data collection would be helpful to contextualise the study.

Further information about the sampled calls is needed: what is their average duration (and how does this compare to calls in general)? Did the researchers draw their sample of 25 calls from food-related calls only, or from calls seeking assistance in other areas too? What drove this decision? Did the 25 sampled calls cover only food-related topics, or were other areas of need also discussed? (this feels especially important in light of the analytical theme of the complex social environment and the competing demands shared by callers). Overall in this section I was left feeling very unclear about exactly how the calls were sampled, and why.

Why did the researchers use thematic analysis? This decision needs justifying.

What is meant by ‘to roll up the codes into themes’? (p4) Clarity is needed here.

Results

Given the use of a purposive sample, there is a circularity in reporting descriptive statistics (because the composition of ZIP codes for example was selected by the researcher) that needs acknowledging.

This is not stated explicitly but the information contained in Table 2 implies that some data was missing (eg: age was reported by 8 callers out of 25). A brief account of why some data are missing and reflection on the possible reasons and implications for this is needed.

In Table 2, what is meant by ‘Household member’? Do you mean household size? If so it would be clearer to state this directly.

Theme 1: Information seeking seems conceptually confused. The quotes listed on page 5 (lines 128-141) actually seem to focus on two themes: information seeking, and callers being referred. These are quite different pathways to seeking help via the 211 helpline and this diversity is not captured in the theme.

Some of the quotes listed on p5-6 (focussing on the theme ‘lack of food and informational barriers’) would more appropriately be coded within the second theme of food insecurity. This overlap between themes is concerning and suggests that further analytical work is needed.

On p6 the authors state ‘In several calls, there is much evidence that callers’ needs have been met’. Who have their needs been met by? Again, for international readers this sort of contextual information is needed.

Some of the quotes included in theme 2 (food insecurity) also overlap with theme 3 (complex social environment). This comment applies particularly to quotes on the theme of competing financial demands. As above, the overlap between themes suggests that further analytical work is needed to identify relevant themes (or at the very least, this fluidity of themes needs acknowledging and reflecting upon).

Discussion

The authors state ‘47% of low-income people do not use food pantries due to informational barriers (i.e., lack of information) [25]. Analyzing the transcripts also highlighted the fact that some people are not using government assistance programs like SNAP but they still have unmet food needs’, implying that lack of information might explain why some callers were not receiving SNAP. Yet the analyses highlighted eligibility issues, not lack of information, as a barrier to SNAP use in this sample. This mismatch between the published research and the findings presented here suggest a limited understanding of the study’s findings, implications, and relevance to existing scholarship on the topic.

The following sentence (p9) does not make sense: ‘Paying for medical bills, unemployment, disability, increasing costs of gas, grocery, and rent, and the impact of COVID-19 on callers’ mental health are among stressors mentioned by callers that have created an unmet food need a complex problem to address’.

Limitations and future work

The authors note that the sample is representative of the Utah population. On what grounds are they making this statement? With my limited understanding of the 211 community helpline I would expect that the sample is one of low-income, vulnerable people, so not representative of the state population at all. If the authors mean that the representative of the urban/rural make up of the Utah population then they ought to state this instead. It would also be helpful to contextualise the findings in relation to the US more generally to help readers appreciate the relevance of this study beyond the immediate study location.

Conclusion

The statement ‘Our analyses captured discrete information that was hidden in food-related information seeking calls’ is confusing. In what way was this information hidden? I would encourage the authors to rephrase this assertion.

Reviewer #2: The qualitative analysis reported in this paper provides a clear answer to a simple and important question: Are calls to a community helpline for food assistance a good proxy for food insecurity? Because the community helpline, 211, is nationally available and widely used, this finding has practical implications for surveillance of food insecurity.

More detail is needed about the sampling approach and inclusion criteria that guided selection of the 211 calls that were analyzed. For example, were there quotas for different demographic sub-groups? Was sampling meant to be proportional to some populations (e.g., 211 callers, residents of Salt Lake County)? Were only English-speaking callers eligible? Were the included transcripts from people who called 211 about food assistance, or from people who called about something else (e.g., utility payment assistance) and were then asked about food?

What is the service area covered by the 211 -- is it all of Utah, or only Salt Lake County? Demographically, those would be quite different.

How long (in word count) were the transcripts? Please report a range and mean. What is meant by "diagram" in Phase 3 of the analysis? What does the diagram visually illustrate? In Phase 4, "vetted" for what?

Can you provide a sense of the relative frequency of each theme? For example, were some more common than others? Also, how much overlap was there among themes? In other words, presumably one transcript could have elements of multiple themes. Did some themes overlap more than others?

Why did so few callers provide age and education information? What is meant by "household member"? Is this a count of household size? Why did so few callers provide this? Were these demographic variables systematically assessed? Inferred from call content?

6. PLOS authors have the option to publish the peer review history of their article (what does this mean?). If published, this will include your full peer review and any attached files.

Reviewer #1: **Yes: **Elisabeth Garratt

Reviewer #2: No

---

## [Author Response · Author response to Decision Letter 0]

24 Jan 2023

Please see the "Response to Reviewers" letter.

---

## [Decision Letter · Decision Letter 1]

7 Mar 2023

PONE-D-22-22201R1

Indications of food insecurity in the content of telephone calls to a community referral system

PLOS ONE

Dear Dr. Sharareh,

Thank you for submitting your manuscript to PLOS ONE. I have now received the reviewer(s) comments for your revised manuscript. After careful consideration, we feel that it has merit but does not fully meet PLOS ONE’s publication criteria as it currently stands. Therefore, we invite you to submit a revised version of the manuscript that addresses the points raised during the review process. While I appreciate your efforts to address the reviewers' initial comments, we are unable to proceed with considering your manuscript for publication without your attention to the comments raised by the reviewer(s) and editor found below. I strongly believe that attention to these issues will greatly improve the manuscript.

**Editor's Comments**

Specifically, in addition to responding to the reviewer's comments below, kindly address or respond to the following:

1. Provide a justification both for the **appropriateness and relevance **of your chosen qualitative analytical approach (thematic analysis) to your research questions/objectives.

2. Overall, I share the reviewer's sentiment about the lack of detail in the analysis, specifically as it relates to theme development, and recommend a more detailed analysis and reporting following conventional qualitative research principles. For example, for theme 1, you indicated that "we selected quotes...". This phrasing speaks to an apriori determination of themes rather than an emergent analytical process that qualitative research strives on. In addition, the narrative accompanying each theme is relatively sparse and descriptive, lacking detailed analysis. Please also pay attention the reviewer #1's original concern about overlapping themes.

3. With respect to sample size determination, please rephrase your methods section to clarify (assuming my understanding is correct) that you obtained an initial sample of 25 transcripts from your partners, completed the analysis, and made a determination on the appropriateness of your sample size based on thematic saturation. Your discussion about this in the results (lines 142-144) should be moved to the methods section.

4. Along these same lines (#3 above), I think it helps the reader appreciate your process more if you clarify that this is a secondary qualitative analysis of existing administrative data (again, assuming my understanding is correct). This will help alleviate some concerns (as raised by Reviewer 1) about your sampling approach. It will also be helpful if you provide justification for this study design in your methods

5. I agree with the reviewer that the conclusion must be clarified. For example, it is unclear to me what you is meant by "hidden in the contents of food-related information..."

6. Finally, please update the revised manuscript to reflect your efforts to fully address all reviewers' specific comments (both for the previous (i.e., revision 1) and current revisions). 

We look forward to receiving your revised manuscript.

Kind regards,

Bettye A. Apenteng

Academic Editor

PLOS ONE

Reviewers' comments:

Reviewer's Responses to Questions

**Comments to the Author**

1. If the authors have adequately addressed your comments raised in a previous round of review and you feel that this manuscript is now acceptable for publication, you may indicate that here to bypass the “Comments to the Author” section, enter your conflict of interest statement in the “Confidential to Editor” section, and submit your "Accept" recommendation.

Reviewer #1: (No Response)

2. Is the manuscript technically sound, and do the data support the conclusions?

Reviewer #1: No

3. Has the statistical analysis been performed appropriately and rigorously? 

Reviewer #1: N/A

4. Have the authors made all data underlying the findings in their manuscript fully available?

Reviewer #1: Yes

5. Is the manuscript presented in an intelligible fashion and written in standard English?

Reviewer #1: Yes

6. Review Comments to the Author

Reviewer #1: Indications of food insecurity in the content of telephone calls to a community referral system

PONE-D-22-22201R1

Second review of manuscript, March 2023

The revised paper shows significant improvements in terms of clarifying some key details of the project’s design and methods. However I still stand by my original recommendation that the paper lacks scholarly rigour and as such does not represent a significant or robust contribution to scholarship.

Primarily I am concerned about the way that calls to the 211 service were sampled. In my original review I asked for clarity on a number of points relating to the selection of phone calls for analysis. The author response states that calls were chosen by their non-academic partner and that the research team did not have access to call recordings. The author is therefore stating that they were not responsible for sampling the phone calls which form the primary data for their analyses. I acknowledge that researchers undertaking secondary data analysis will inevitably face constraints associated with this method but the lack of involvement of trained researchers in sampling for this project is a fatal flaw in terms of data quality and academic rigour more generally.

I also have further concerns about the paper’s scholarly rigour:

• In the abstract the authors quantify the sample composition (‘There were 13 calls from metropolitan and 12 calls from nonmetropolitan ZIP Codes. Participants included 5 Hispanic, 8 non-Hispanic, 7 white, and 5 non-whites.), but this is not a meaningful approach when undertaking qual research. Given that purposive sampling was used to determine the sample composition, it is an entirely circular statement that demonstrates a poor understanding of qualitative research approaches

• The authors justify thematic analysis on the basis of its popularity, not in relation to its appropriateness to exploring their research questions. Having been prompted to provide clear justification for this decision, their failure to do so is indicative of a careless approach to research design

• The authors make assertions that are not supported by the data they report. For example, when I queried their statement that ‘In several calls, there is much evidence that callers’ needs have been met’, the revised paper now reads ‘In several calls, there is much evidence that callers’ information needs have been met by the operator’, without providing any data to support this assertion.

• In several places, the authors (apparent) response to reviewer comments is missing from the revised paper. For example, I questioned the meaning of the sentence ‘Our analyses captured discrete information that was hidden in food-related information seeking calls’ and the authors made reference to a revised sentence on line 334. The revised manuscript does not have a line 334 and a text search of the term ‘hidden’ does not reveal any changes to the manuscript. This is either the mark of careless scholarship or a deliberate attempt to mislead the reviewers and editor; either option is very discouraging

• More generally, several author responses to the reviewer comments have not been incorporated into the manuscript, suggesting that the authors have poor knowledge of the purpose of peer review

7. PLOS authors have the option to publish the peer review history of their article (what does this mean?). If published, this will include your full peer review and any attached files.

Reviewer #1: **Yes: **Elisabeth Garratt

---

## [Author Response · Author response to Decision Letter 1]

27 Mar 2023

Thank you for the opportunity to revise and resubmit our manuscript (PONE-D-22-22201R1), entitled “Indications of food insecurity in the content of telephone calls to a community referral system” for reconsideration by PLOS ONE. 

We thank the editor and reviewer for their useful comments. We appreciate the opportunity to address the comments by submitting a revision. We have considered all comments and believe that the revisions have addressed these comments and greatly improved the paper.

We realize that reviewer 1 likely did not have access to the marked-up version of the manuscript in our first resubmission and this may have contributed to some confusion. We have taken care to be clearer about each modification in response to her comments for resubmission #2. 

Below, we have addressed the editor’s and reviewer’s concerns by explaining in detail the responses (in bold) to their questions/concerns (numbered). We have highlighted the changes in the manuscript with the “track changes” function. We have now provided quotes in addition to line numbers in case these are not visible to the reviewer. Line numbers refer to the “Revised Manuscript with Track Changes” document.

Editor

1. Provide a justification both for the appropriateness and relevance of your chosen qualitative analytical approach (thematic analysis) to your research questions/objectives.

Please see the “Qualitative Analysis Approach” section and below:

We used thematic analysis to analyze our unconventional but novel qualitative data—transcripts of food-related calls. Thematic analysis is a powerful approach to determining a set of experiences, thoughts, or behaviors across qualitative data and provides a well-structured approach to assessing the perspectives of research participants and highlighting their similarities and differences. This approach organizes qualitative data into codes and themes, similar to other qualitative methodologies such as grounded theory and ethnography. Thematic analysis is also widely used in qualitative analysis and in health-related fields and works well at bringing together multidisciplinary teams.

2. Overall, I share the reviewer's sentiment about the lack of detail in the analysis, specifically as it relates to theme development, and recommend a more detailed analysis and reporting following conventional qualitative research principles. For example, for theme 1, you indicated that "we selected quotes...". This phrasing speaks to an apriori determination of themes rather than an emergent analytical process that qualitative research strives on. In addition, the narrative accompanying each theme is relatively sparse and descriptive, lacking detailed analysis. Please also pay attention the reviewer #1's original concern about overlapping themes.

We appreciate the editor and reviewer’s concerns about the lack of analytic detail. We have made adjustments to the manuscript language to make our process clearer. Throughout our process, we have used code-based thematic groupings, meaning that quotes were assigned to codes and those codes were grouped into themes that we then described at a higher level of synthesis.

However, given the concern about overlapping themes (reviewer #1’s original concern), we revisited our analysis and changed our thematic structure to represent three overarching themes. These themes are (1) referral to 211; (2) reasons for food-related calls; and (3) reasons for unmet food needs. We present sample quotes for each theme in the manuscript. We believe that these 3 themes assist in highlighting what is unique about each theme.

3. With respect to sample size determination, please rephrase your methods section to clarify (assuming my understanding is correct) that you obtained an initial sample of 25 transcripts from your partners, completed the analysis, and made a determination on the appropriateness of your sample size based on thematic saturation. Your discussion about this in the results (lines 142-144) should be moved to the methods section.

Thank you for your suggestion. We moved that discussion to the Method section, Lines 146-148. 

Please also see our response to reviewer #1’s concern, which provide details on our sampling approach.

4. Along these same lines (#3 above), I think it helps the reader appreciate your process more if you clarify that this is a secondary qualitative analysis of existing administrative data (again, assuming my understanding is correct). This will help alleviate some concerns (as raised by Reviewer 1) about your sampling approach. It will also be helpful if you provide justification for this study design in your methods.

We appreciate your suggestion and added this point in the methods section, Line 100.

Please also see our response to reviewer #1’s concern and the new “Qualitative Analysis Approach” section.

5. I agree with the reviewer that the conclusion must be clarified. For example, it is unclear to me what you meant by "hidden in the contents of food-related information..."

We revised the conclusion section and now it reads as follow:

Our secondary qualitative analyses of the transcripts of food-related information seeking calls revealed information that could be critical to policymaking and the allocation of food resources. Specifically, we highlighted the relevance of 211 food-related calls to food insecurity and identified the complex social environment and multifaceted barriers that callers are dealing with. Indications of food insecurity through these calls validate the use of these calls as a proxy measure for food insecurity, however, further analysis is needed to measure FI among these callers using established methods. We also highlighted the need for educational campaigns to increase awareness about available food resources.

6. Finally, please update the revised manuscript to reflect your efforts to fully address all reviewers' specific comments (both for the previous (i.e., revision 1) and current revisions). 

We uploaded two versions. One is the marked-up version with track changes, which highlights the revisions we made in the second resubmission. The other one is the unmarked version, called “manuscript”. 

In the “Response to Reviewers” letter, we referred you and the review to different line numbers based on the marked-up version. Please also see our response to reviewer #6’s concern, in which, we provide the line numbers of our responses to the original concerns.

Reviewer #1 (Elisabeth Garratt)

1. Primarily, I am concerned about the way that calls to the 211 service were sampled. In my original review, I asked for clarity on a number of points relating to the selection of phone calls for analysis. The author response states that calls were chosen by their non-academic partner and that the research team did not have access to call recordings. The author is therefore stating that they were not responsible for sampling the phone calls which form the primary data for their analyses. I acknowledge that researchers undertaking secondary data analysis will inevitably face constraints associated with this method but the lack of involvement of trained researchers in sampling for this project is a fatal flaw in terms of data quality and academic rigour more generally. 

Thank you for raising this concern. Below and in the manuscript (Methods section), we clarified the sampling process.

We conducted a secondary qualitative analysis of existing administrative data. The existing data were audio recordings of food-related calls to Utah’s 211. The United Way of Salt Lake keeps audio recordings for a month from the time of the call. While we did not have access to the audio recordings, we provided the United Way of Salt Lake with a sampling algorithm, with one main criteria, to identify 25 food-related calls and transcribe them for our research team. We initially requested 25 transcripts as it is a reasonable sample size for qualitative analysis. Our main criterion was to receive transcripts of 13 calls from metropolitan and 12 calls from nonmetropolitan ZIP Codes to ensure representation of rural residents as they might have different food insecurity experiences. In Utah, 47% of ZIP Codes are nonmetropolitan areas according to the rural-urban commuting area codes. Moreover, using a purposive sampling approach, we requested a sample that varied in regard to race and ethnicity similar to the Utah population. 77% of Utah's population are non-Hispanic white and 14.4% of the population are Hispanic or Latino. Considering that Utah 211 collects race/ethnicity in the same field, non-Hispanic could be white or any other race and non-white may be Hispanic. We also did not restrict our sample to English-speaking callers, but we do not know whether those callers used a non-English operator or a translation line as we were only provided the transcript of calls. From February to March 2022, the 211 senior community resource specialist identified the food-related calls that matched our criteria upon receiving them, transcribed the calls within the next few days, and provided us with 25 de-identified transcripts. While we did not select those calls, we designed the sampling criteria and made sure that our final sample was aligned with those criteria.

2. In the abstract the authors quantify the sample composition (‘There were 13 calls from metropolitan and 12 calls from nonmetropolitan ZIP Codes. Participants included 5 Hispanic, 8 non-Hispanic, 7 white, and 5 non-whites.), but this is not a meaningful approach when undertaking qual research. Given that purposive sampling was used to determine the sample composition, it is an entirely circular statement that demonstrates a poor understanding of qualitative research approaches.

We clarified our sampling strategy in the manuscript and specified the numbers in the abstract’s methods section. 

3. The authors justify thematic analysis on the basis of its popularity, not in relation to its appropriateness to exploring their research questions. Having been prompted to provide clear justification for this decision, their failure to do so is indicative of a careless approach to research design.

We apologize for the lack of details in our prior response. We added more details to the manuscript to justify our choice of method. Please see the “Qualitative Analysis Approach” section and below:

We used thematic analysis to analyze our unconventional but novel qualitative data—transcripts of food-related calls. Thematic analysis is a powerful approach to determining a set of experiences, thoughts, or behaviors across qualitative data and provides a well-structured approach to assessing the perspectives of research participants and highlighting their similarities and differences. This approach organizes qualitative data into codes and themes, similar to other qualitative methodologies such as grounded theory and ethnography. Thematic analysis is also widely used in qualitative analysis and in health-related fields and works well at bringing together multidisciplinary teams.

4. The authors make assertions that are not supported by the data they report. For example, when I queried their statement that ‘In several calls, there is much evidence that callers’ needs have been met’, the revised paper now reads ‘In several calls, there is much evidence that callers’ information needs have been met by the operator’, without providing any data to support this assertion.

Thank you for pointing out to this problem. The quotes are now removed as they do not fall into our informative selected themes.

5. In several places, the authors (apparent) response to reviewer comments is missing from the revised paper. For example, I questioned the meaning of the sentence ‘Our analyses captured discrete information that was hidden in food-related information seeking calls’ and the authors made reference to a revised sentence on line 334. The revised manuscript does not have a line 334 and a text search of the term ‘hidden’ does not reveal any changes to the manuscript. This is either the mark of careless scholarship or a deliberate attempt to mislead the reviewers and editor; either option is very discouraging.

We appreciate your concern and we apologize for any misunderstanding. Below, please see the milestone for our revisions, which indicate that we had no intention for misleading.

• In the first submission, this sentence was: “Our analyses captured discrete information that was hidden in food-related information seeking calls and would be critical to informing policymakers”.

• You raised this concern: “In what way was this information hidden?”

• Therefore, in the first resubmission, we revised this sentence by adding the word “content” to indicate that the information was hidden in the content of transcripts: “Our analyses captured information that was hidden in the contents of food-related information seeking calls and would be critical to informing policymakers”. However, we might have misunderstood your point.

• We also referred the reviewers to line 334 based on the “Revised Manuscript with Track Changes” document. You might have looked at the version without track changes or did not have access to the document with track changes.

In this second resubmission, we revised this sentence. Please see the conclusion section.

Please also note that when we refer the reviewers to a line number, it is based on the version that has track changes on—the marked-up version.

6. More generally, several author responses to the reviewer comments have not been incorporated into the manuscript, suggesting that the authors have poor knowledge of the purpose of peer review.

We believe that we had addressed the majority of your concerns and incorporated them into the manuscript. In a few instances, we provided explanations and did not incorporate them into the manuscript; but now, everything is in there.

As requested by the editor, we have provided line numbers for our responses to your previous concerns to facilitate the review process. These line numbers are based on the “Revised Manuscript with Track Changes” document.

Review 1’s concerns from the previous submission

1. While interesting enough, I am not recommending this paper to be accepted for publication in PLOS One, for two key reasons. 

a. First, by reporting on a small study with a specific geographical context, the scope of its contribution is small, and insufficient to merit inclusion in an international journal such as PLOS One. When reworking this paper for submission elsewhere I would encourage the authors to think carefully about how their findings might have relevance beyond the immediate research context and bring this out more clearly in the revised work. 

b. Second, I have reservations about the paper’s scholarly rigor, especially the analyses, which identified overlapping analytical themes and thereby felt weak and lacking robustness.

In the second resubmission, we have clarified the sampling approach (starting Line 100), explained the rationale for using thematic analysis (Qualitative Analysis Approach section), and provided more contextual information about the 211 network (introduction section, lines 68-84).

We have also revised our themes and provided 3 new themes that do not overlap with each other (please see the results section). 

2. The title ‘Real-time indications of food insecurity in telephone calls to a community referral system’ is not an accurate representation of the focus of this paper, as the paper contains no temporal element (such as how calls to the community referral system have changed over a given time period, or fluctuations over the course of a month). I would like to see the title provide a closer reflection of the paper’s focus and scope.

Please our new title: “Indications of food insecurity in the content of telephone calls to a community referral system”.

3. Linking natural disasters to food security (p2) is an important consideration in the Global South but its relevance to the Global North (where this study is situated) is not clear.

Please see lines 55-56.

4. The sentence ‘Lack of donations to food pantries [7] and the increasing rate of inflation [8] make it harder for food pantries to meet the demand and drive them to closure, which will put people who rely on emergency food resources at risk of food insecurity’ (p2) misunderstands the nature of food insecurity and the role of food banks in assisting those experiencing food insecurity. People who rely on emergency food are almost by definition experiencing food insecurity, and the inadequacy of food banks in alleviating food insecurity is well documented. It is therefore inaccurate to talk of food banks as protecting people from food insecurity.

Please see lines 60-61.

5. More contextual information is needed about the 211 network (p3). In particular, what is its purpose? Who is it funded by? Does it offer solely information or is it a means of callers being referred to relevant services? And who are United Way? Your international readers may not have heard of this organisation or be familiar with its goals and purpose of operation.

Please see lines 68-84.

6. The authors state that ‘Researchers have also used food-related calls to 211 as a proxy for food insecurity’. What is the value for doing so? What can it add to national prevalence estimates, for example? Unpacking this point would help the authors demonstrate the value of this study more convincingly.

Please see lines 86-88 (last paragraph of the Introduction section).

7. The justification for locating this study in Utah need justifying. Presumably it reflects the researchers’ location but a more convincing reason is needed, alongside reflection on the potential transferability of the project findings to other settings.

This manuscript stemmed from a unique community partnership that the first author had with the Utah’s 211 system (Please see line 95). 

Please also see the limitation section, Lines 395-399.

8. Why did the researchers sample 25 calls? What proportion of monthly calls to the 211 helpline is this?

25 transcripts were chosen as it is a reasonable sample size for qualitative analysis. Please see line 105.

After analyzing these 25 transcripts, we did not see the emergence of new themes and felt comfortable that we had reached saturation. Please see line 146.

9. Dates of data collection would be helpful to contextualize the study.

Please see line 114.

10. Further information about the sampled calls is needed: what is their average duration (and how does this compare to calls in general)? Did the researchers draw their sample of 25 calls from food-related calls only, or from calls seeking assistance in other areas too? What drove this decision? Did the 25 sampled calls cover only food-related topics, or were other areas of need also discussed? (this feels especially important in light of the analytical theme of the complex social environment and the competing demands shared by callers). Overall in this section I was left feeling very unclear about exactly how the calls were sampled, and why.

Please see Table 2, lines 117-120, and the updated sampling approach in the methods section.

11. Why did the researchers use thematic analysis? This decision needs justifying.

Please see the “Qualitative Analysis Approach” section.

12. What is meant by ‘to roll up the codes into themes’? (p4) Clarity is needed here.

We revised this sentence and now it reads as “to create a diagram to illustrate and organize codes into themes.” Please see lines 141-142.

13. Given the use of a purposive sample, there is a circularity in reporting descriptive statistics (because the composition of ZIP codes for example was selected by the researcher) that needs acknowledging.

Please see the updated Table 2 caption and the footnote under Table 2, and the updated abstract’s method section.

14. This is not stated explicitly but the information contained in Table 2 implies that some data was missing (eg: age was reported by 8 callers out of 25). A brief account of why some data are missing and reflection on the possible reasons and implications for this is needed.

Please see lines 159-161.

15. In Table 2, what is meant by ‘Household member’? Do you mean household size? If so it would be clearer to state this directly.

We changed that to household size.

16. Theme 1: Information seeking seems conceptually confused. The quotes listed on page 5 (lines 128-141) actually seem to focus on two themes: information seeking, and callers being referred. These are quite different pathways to seeking help via the 211 helpline and this diversity is not captured in the theme.

Please see our new themes.

17. a. Some of the quotes listed on p5-6 (focussing on the theme ‘lack of food and informational barriers’) would more appropriately be coded within the second theme of food insecurity. This overlap between themes is concerning and suggests that further analytical work is needed. Some of the quotes included in theme 2 (food insecurity) also overlap with theme 3 (complex social environment). This comment applies particularly to quotes on the theme of competing financial demands. As above, the overlap between themes suggests that further analytical work is needed to identify relevant themes (or at the very least, this fluidity of themes needs acknowledging and reflecting upon).

Please see our new themes.

b. On p6 the authors state ‘In several calls, there is much evidence that callers’ needs have been met’. Who have their needs been met by? Again, for international readers this sort of contextual information is needed.

This sentence is now removed.

18. The authors state ‘47% of low-income people do not use food pantries due to informational barriers (i.e., lack of information) [25]. Analyzing the transcripts also highlighted the fact that some people are not using government assistance programs like SNAP but they still have unmet food needs’, implying that lack of information might explain why some callers were not receiving SNAP. Yet the analyses highlighted eligibility issues, not lack of information, as a barrier to SNAP use in this sample. This mismatch between the published research and the findings presented here suggest a limited understanding of the study’s findings, implications, and relevance to existing scholarship on the topic.

While only one caller mentioned the ineligibility for SNAP (as shown in Theme 3), many other callers that had food needs were not using any government assistance programs including SNAP (see Table 2). The 211 specialists sometimes ask a question about using government assistance programs, however, this is not a common practice. 

19. The following sentence (p9) does not make sense: ‘Paying for medical bills, unemployment, disability, increasing costs of gas, grocery, and rent, and the impact of COVID-19 on callers’ mental health are among stressors mentioned by callers that have created an unmet food need a complex problem to address’.

Please see lines 277-280.

20. The authors note that the sample is representative of the Utah population. On what grounds are they making this statement? With my limited understanding of the 211 community helpline I would expect that the sample is one of low-income, vulnerable people, so not representative of the state population at all. If the authors mean that the representative of the urban/rural make up of the Utah population then they ought to state this instead. It would also be helpful to contextualise the findings in relation to the US more generally to help readers appreciate the relevance of this study beyond the immediate study location.

Please see the revised limitations section.

21. The statement ‘Our analyses captured discrete information that was hidden in food-related information seeking calls’ is confusing. In what way was this information hidden? I would encourage the authors to rephrase this assertion.

Please see the revised conclusion section.

---

## [Editor Report · Decision Letter 2]

5 Apr 2023

Indications of food insecurity in the content of telephone calls to a community referral system

PONE-D-22-22201R2

Dear Dr. Sharareh,

We’re pleased to inform you that your manuscript has been judged scientifically suitable for publication and will be formally accepted for publication once it meets all outstanding technical requirements.

Kind regards,

Bettye A. Apenteng

Academic Editor

PLOS ONE

---

## [Editor Report · Acceptance letter]

11 Apr 2023

PONE-D-22-22201R2 

Indications of food insecurity in the content of telephone calls to a community referral system 

Dear Dr. Sharareh:

I'm pleased to inform you that your manuscript has been deemed suitable for publication in PLOS ONE. Congratulations! Your manuscript is now with our production department. 

Kind regards, 

on behalf of

Dr. Bettye A. Apenteng 

Academic Editor

PLOS ONE